# The Microfluidic Toolbox for Analyzing Exosome Biomarkers of Aging

**DOI:** 10.3390/molecules26030535

**Published:** 2021-01-20

**Authors:** Jonalyn DeCastro, Joshua Littig, Peichi Peggy Chou, Jada Mack-Onyeike, Amrita Srinivasan, Michael J. Conboy, Irina M. Conboy, Kiana Aran

**Affiliations:** 1Keck Graduate Institute, The Claremont Colleges, Claremont, CA 91711, USA; jherce@kgi.edu (J.D.); jlittig18@students.kgi.edu (J.L.); jmack18@students.kgi.edu (J.M.-O.); 2Pitzer College, The Claremont Colleges, Claremont, CA 91711, USA; pechou@students.pitzer.edu; 3Claremont McKenna College, The Claremont Colleges, Claremont, CA 91711, USA; asrinivasan23@students.claremontmckenna.edu; 4Department of Bioengineering, University of California, Berkeley, CA 94720, USA; conboymj@berkeley.edu (M.J.C.); iconboy@berkeley.edu (I.M.C.)

**Keywords:** exosomes, aging, microfluidics

## Abstract

As the fields of aging and neurological disease expand to liquid biopsies, there is a need to identify informative biomarkers for the diagnosis of neurodegeneration and other age-related disorders such as cancers. A means of high-throughput screening of biomolecules relevant to aging can facilitate this discovery in complex biofluids, such as blood. Exosomes, the smallest of extracellular vesicles, are found in many biofluids and, in recent years, have been found to be excellent candidates as liquid biopsy biomarkers due to their participation in intercellular communication and various pathologies such as cancer metastasis. Recently, exosomes have emerged as novel biomarkers for age-related diseases. Hence, the study of exosomes, their protein and genetic cargo can serve as early biomarkers for age-associated pathologies, especially neurodegenerative diseases. However, a disadvantage of exosome studies includes a lack in standardization of isolating, detecting, and profiling exosomes for downstream analysis. In this review, we will address current techniques for high-throughput isolation and detection of exosomes through various microfluidic and biosensing strategies and how they may be adapted for the detection of biomarkers of age-associated disorders.

## 1. Introduction

Aging is the highest risk factor in developing numerous degenerative diseases, such as Alzheimer’s and cardiovascular diseases, or developing various cancers [1]. Currently, biomedical advances are in progress to extend health and life while simultaneously combatting the degenerative and inflammatory-based diseases that arise with aging. Recently, these advances include research and clinical applications with extracellular vesicles. Exosomes, the smallest of extracellular vesicles, have been implicated in cell-to-cell communication and transmission of “young” to “aged” cell state [2].

Exosome participation in cell-to-cell communication and their biocompatible nature makes them invaluable biomarker candidates for diagnostics and liquid biopsies. While other biomarkers such as circulating tumor cells (CTCs), circulating cell-free DNA (ccf-DNA), and soluble proteins are more readily metabolized within the blood, exosomes can resist degradation of important biomarkers by embedding proteins within their plasma membrane or encapsulating proteins, different RNA types, and DNA internally [3,4,5,6]. This is especially true for cancer states, where tumor cells secrete exosomes into the bloodstream, which are detectable at a higher concentration than CTCs and ccf-DNA. In addition, several studies have shown that the increased secretion of specific subtypes of exosomes into the blood is associated with some pathologies [3,4,6]. Therefore, exosomes have significant potential to serve as early liquid biopsy biomarkers for many pathologies. However, a universal bottleneck of exosome research includes the extraneous means of isolation and detection given the current conventional methods such as ultracentrifugation or density gradient centrifugation are both time-consuming and result in a variable or low yield [7]. Furthermore, there is no standardized detection method for exosomes where exosome confirmation is only through a combination of methods to avoid any purification bias [8]. Additionally, there is a high degree of heterogeneity among exosome populations, given that any cell can secrete exosomes. Figure 1 shows a plethora of surface and intravesicular markers that can vary for a single exosome [9]. Generally, each exosome can have varying distributions of any one of these markers. Given these considerations, confirmation of exosomes may take place through a combination of size, surface or internal marker confirmation, and vesicle morphology verification. Ultimately, this extensive isolation and detection phase for exosome research impedes those who are trying to enter these forthcoming exosome fields.

To address these challenges, a substantial amount of research interest is focused on microfluidic tools to improve the throughput and consistency of exosome research in recent years. Some of the more successful strategies have already been applied to both cancer and neurological fields. In this article, we will first explore the current microfluidic tools that are streamlining the isolation and detection of exosomes and propose the current topics of aging, more specifically, among cancer and neurodegenerative disease research that may benefit from these tools. Additionally, we will explore emerging microfluidic techniques for monitoring the mechanisms at which exosomes play a role in neurodegenerative transmission.

## 2. The Potential of Exosomal Biomarkers for Precision Medicine and Liquid Biopsies

Extracellular vesicles are categorized by their size and mode of secretion (Figure 2) [10,11]. The smallest of these exosomes have a diameter range of 30–150 nm and are formed through calcium-mediated endocytosis pathways. Exosome formation begins with an early endosome, which advances to a multivesicular body (MVB). The MVB holds one of two fates: (1) directed toward a lysosome for digestion or (2) for exocytosis out of the cell (Figure 2) [10,12,13,14]. This pathway is one of the ways that exosomes set themselves apart from their larger counterparts since exosome endosomal formation can intersect with vesicle formation from cell organelles. Through this pathway, exosomes can carry information about cell organelles and plasma membrane biomarkers. Emerging evidence shows that exosomes are deliberately packaged and carry valuable information from their parent cell. Exosome cargo is in the form of DNA, RNA, metabolic enzymes (peroxidases, lipid kinases), tetraspanins (CD9, CD63, CD81), heat shock proteins (HSP70, HSP90), and cytosolic proteins (tubulin, actin)—all specific to parent cell function (Figure 1) [10,11,12,13,14]. Various exosome surface biomarkers and cargo provide information about their originating parent cells as well as intended target cells; thus, exosomes provide information about their intended function, making them excellent biomarkers for precision-medicine and liquid biopsies.

## 3. Microfluidic Solutions for Exosome Isolations

The time-consuming and extensive measures needed for isolation and detection of exosomes impedes the progress of exosome research. Exosome isolation from blood is the first critical step in utilizing them as biomarkers for liquid biopsy. During exosome isolation, exosomes can be mistaken as proteins, lipid aggregates or other cell debris due to their small diameters. Size-based conventional methods to aid with appropriate exosome isolation include ultracentrifugation (UC), filtration or ultrafiltration (UF), or size-exclusion chromatography (SEC) (Appendix A). However, these methods are laborious and, in the case of UC or UF, require large and expensive equipment. Furthermore, the high centrifugal speeds required for UC can yield co-precipitated protein aggregate contamination [7,15]. Another conventional isolation technique includes precipitation, of which many commercially made kits such as Exo-spin, ExoQuick, miRCURY exosome isolation kit utilize. Within these commercial kits, biofluids are exposed to a polymer matrix, which causes exosome precipitation [16,17]. These kits are useful in that no additional equipment is needed outside of a normal benchtop centrifuge; however, it is unclear whether the exposure of exosomes to polymers may influence their properties [7,18]. A popular approach for conventional exosome isolation and detection includes immunoaffinity-based approaches. However, a disadvantage of using this method alone is that given the secreting cell, other extracellular vesicles outside of pure exosomes may have the same surface markers [7,11]. A full comparison of the advantages and disadvantages of conventional methods for exosome isolation and detection are provided in Appendix A.

To address the current disadvantages of conventional exosomal isolation techniques, microfluidic systems have been designed and implemented. Within these systems, microfluidic tools utilize the principles of conventional techniques but scale them down to micro or nano functionality enabling smaller sample volumes and adaptation to the point-of-care setting.

### 3.1. Field-Based Isolation of Exosomes

Recent studies of microfluidic-based tools for the isolation of exosomes show popularity in using field-based methods such as applying an electric, acoustic, or magnetic field for separation. Some of these techniques additionally combine the use of filtration or labeling for further exosome isolation and characterization. Studies show that utilizing these techniques allow for minimal pre-sample processing from biofluids, and potential for single-step, on-chip isolation within a fraction of the time of conventional methods.

Wu et al. integrate acoustics and microfluidics, allowing for the label-free isolation of exosomes. This acoustofluidic platform has two modules: a microscale cell-removal module to remove larger blood components and an exosome isolation module for extracellular vesicle (EV) subgroup separation, integrated together into a single acoustofluidic device [12]. Interestingly, the principles behind the cell and exosome separation are based on sequential surface acoustic wave modules where the periodic distribution of acoustic on and off modes results in an acoustic radiation force that pushes larger particles and exosomes toward separate node and antinode planes. The device itself appeared to be useful for a range of biofluids. However, one challenge was realized when processing blood with a high level of complexity between blood lipid levels. The aggregation of lipid particles may disrupt both the laminar flow and the acoustic field and affect the device’s separation capability. Wu et al. proposed adjusting the sheath/sample flow when testing blood samples with high lipid levels and/or adding an additional isolation modulation designed specifically for lipid removal [12].

Zhang et al. 2018 utilized asymmetric flow field-fractionation (AF4) to address the confounding issue of exosome heterogeneity by separating exosomal subtypes along size differences [19]. AF4 uses forward laminar channel flow, in which flow velocity is highest in the center of the channel and lowest along the channel walls, and an externally applied crossflow in a perpendicular flow arrangement pushes sample particles to the accumulation wall, which features an ultrafiltration membrane and a porous bottom plate [20]. This flow counteracts that of the diffusion of the particles away from the wall, resulting in the separation of nanoparticles based on their density and hydrodynamic properties. Using this methodology, they were able to identify three subpopulations of exosomes: large exosomes (90–120 nm), small exosomes (60–80 nm) and ~35 nm diameter nanoparticles that they call “exomeres”. Furthermore, this separation allowed for downstream analysis of these subtypes to have distinct compositions of N-glycosylation, protein, lipid, DNA, RNA, and biophysical properties [19].

Another field-based technique includes surface plasmon resonance (SPR) detection, which is typically used for the detection of nano-molecules by detecting minute changes in the electrical conductance of electrons at the interface between two surfaces upon an incidence of light at the surface [21]. Thakur et al. 2017 gives a proof-of-concept model for localized surface plasmon resonance (LSPR) without the need for immunolabeling or functionalization of the surface to provide a low-cost and less time-consuming alternative for exosome detection. Exosomes and microvesicles (MVs) were isolated and purified. With LSPR, the electromagnetic field induced was confined to the nanosurface of randomly distributed, self-assembled gold nanoislands (SAM-AuNIs) that were fabricated in a two-step deposition-annealing process. This random distribution of gold nanoislands increases the surface area available for exosome interaction. LSPR spectral phase detection was read through interferometry and was utilized to observe exosome adsorption onto the surface [21].

To expand on field-based isolation techniques, Dudani et al. 2015 (Figure 3A) and Cho et al. 2016 (Figure 3B) integrated other standard methods of isolation with an externally applied field to facilitate exosome isolation. Dudani discusses the use of a microfluidic device that incorporates rapid inertial solution exchange (RInSE) for exosome isolation [22]. Biotinylated anti-human CD63 antibodies (specific exosome marker) were conjugated to the streptavidin-coated polystyrene exosome capture beads and injected into the device with tris buffer solution (TBS) as the exchange solution (Figure 3Ai). The sample fluid is presented with a co-flow of TBS sheath fluid at different flow rates (Figure 3Aii). Due to inertial lift forces, the exosome bound beads were streamlined into the exchange buffer, and contaminants were siphoned off [23]. A flow cytometer and an inline fluorescence detection system were integrated into the device, which allowed for integrated downstream analysis of the size and other molecular and protein analysis. On the other hand, Cho coupled electrophoretic migration with size-exclusion filtration. The Cho device actuates nanoparticle separation based on zeta-potential, which is proportional to particle size [24]. Exosomes and most proteins have a negative surface charge and zeta potential. The device utilizes a two-electrode and two-nanoporous (30 nm) membrane system in a sandwiched configuration, which allows the horizontal flow of sample and buffers with a fixed flow rate and vertical migration of particles towards the nanoporous membrane from the applied voltage (Figure 3Bi). Figure 3Bii shows an implementation diagram of the device where negatively charged particles (exosomes and proteins) migrate towards the anode, ensuring interaction with the nanoporous membrane. Proteins are small enough to flow through the filter pores, and exosomes are trapped on the surface of the membrane. All neutral and positively charged particles (impurities) gravitate towards the cathode and filter out.

Although these methods do not encompass the depth of field-based microfluidic tools for exosome screening, this discussion highlights the capabilities of microfluidic techniques in optimizing exosome isolation. These techniques offer low-cost, hand-held devices that make exosomal biofluid screening at the point-of-care possible. Additionally, unlike the conventional assays featured in Appendix A, these methods can isolate exosomes in a fraction of the time, within minimal steps, and include many applications for rapid, downstream analysis. However, some caveats with these tools are that depending on the complexity of the biofluid, unclogging, or sample preprocessing may need to occur for accurate readings. Additionally, specific field-based parameters such as flow rate, wave frequency, or charge distribution may require optimization.

### 3.2. Surface-Based Isolation of Exosomes

Another common microfluidic tool for exosome isolation consists of surface functionalization for the enrichment and capture of exosomes. Recent studies show that antibodies against specific exosome surface markers such as CD9, CD81, or CD63 have been used for high-throughput screening of exosomes. Additionally, studies also incorporate microfluidic tools utilizing aptamers for exosome isolation. These methods can be further integrated with various biosensing materials for exosome detection.

Xia et al. 2017 describe the use of single-walled carbon nanotubes (s-SWCNTs) and an aptamer to detect exosomes. Aptamers can be ideal antigen recognition elements given their structure can be easily modifiable based on pairing with complementary strands, electrostatic interactions, and hydrogen bonding [25]. In some cases, they are even superior to antibody-based recognition based on their chemical stability, higher affinity and potentially higher fixation density on electrode surfaces. An aptamer against CD63 was then adsorbed onto the surface of the s-SWCNTs through non-covalent interactions. The s-SWCNTs catalyzed the oxidation of 3,3′,5,5′-tetramethylbenzidine (TMB) in the presence of H_2_O_2_, resulting in a solution color change from colorless to a moderate blue, measurable through UV-visible (UV-vis) absorption (Figure 4Ai). In contrast, the presence of exosomes caused the de-absorption of the aptamer from the s-SWCNTs, reducing the catalytic activity and causing the solution of TMB to become a lighter blue and a lower measured UV-vis response (Figure 4Aii) [26].

Using a slightly different approach, Chen et al. 2018 developed the ZnO-chip biosensor to detect exosomes by incorporating a size-exclusion-like effect on the standard immunocapture and colorimetric assay. The microfluidic device consists of two parts: ZnO nanowires fit exosome diameter range and a microporous chip embedded with a three-dimensional (3D) polydimethylsiloxane (PDMS) scaffold (Figure 4Bi, ii). Then ZnO nanocrystals were seeded into the scaffold (Figure 4Bi) [27]. The ZnO nanowires were additionally decorated with anti-CD63, and this unique ZnO wire antibody immobilization presented a larger surface area for capturing exosomes (Figure 4Biii). Exosomes were introduced to the ZnO chip via blood serum at an optimum flow rate of 10 uL/min with exosome detection visualized with a similar TMB colorimetric reaction (Figure 4Biv).

The value in these microfluidic colorimetric tools is the ease of substituting the UV-vis detection with a smartphone or computer, which eliminates the requirement of extensive lab equipment for exosome screening. Liang et al. 2017 accomplished this with their integrated double-filtration microfluidic device, which allows for the separation, enrichment, and quantification of exosomes from urine through the succession of the two device parts. As shown in Figure 4Ci, device I utilizes polycarbonate membranes with 200 nm pores, and device II receives device I filtrate for further exosome enrichment with a 30 nm pore-size membrane. When applied to biological samples, urine exosomes were detected using a direct microchip ELISA. The exosomes were labeled with biotinylated anti-CD63 to interact with streptavidin-HRP (Figure 4Cii), and a colorimetric readout with TMB can be read with a smartphone and computer (Figure 4Ciii) [28]. This on-chip reading could allow complex exosome screening to take place within a point-of-care setting and without the need for a full central lab set up.

Another method of combined isolation and detection is surface-based isolation and electrical readouts, which offer an advantage over the previously described optical methods as it allows for real-time measurement. This is advantageous as it allows researchers and clinicians to take immediate action when profiling exosomal biomarkers. Tsang et al. 2019 features a combination of immuno- and electronic-based isolation and detection of exosomes with the speed and selectivity of a graphene-based field-effect transistor (gFET). The gFET biosensor features a monolayer of graphene functionalized with 1-pyrenebutyric acid N-hydroxysuccinimide ester (PBASE) that serves as a heterobifunctional linker [29] (Figure 5A). For exosome detection, anti-CD63 was functionalized to bind to NHS on the surface of the graphene. A microfluidic channel was incorporated into the gFET to allow the exosomes to interact with anti-CD63 immobilized on the surface of the graphene. Binding of the exosomes to the surface of the graphene modulates the electrical properties of the system proportional to the concentration of bound exosomes, which changes the conductivity of the graphene and is readily detected electronically (Figure 5B).

The proposed surface functionalization tools for profiling of exosome biomarkers are some of the most widely used techniques in the field [7]. One major benefit for microfluidic incorporation of antibodies or aptamers on a sensing surface is for simultaneous isolation and detection of exosomes of interest. Furthermore, the mesh or parallel structures of which the surfaces are layered and constructed to offer a unique geometry that not only allows for increased surface area for exosomal binding but also provides an opportunity for multiplexing by utilizing antibodies for different surface markers. In addition, increased exosomal interaction with a sensing surface allows for more accurate quantification of exosome physical characteristics such as size, concentration, population subtype, and correlation with disease specificity, which are discussed in the following section.

## 4. Exosomal Detection Systems to Monitor Age-Associated Pathologies

Recently, there has been an exponential increase in the number of reports regarding exosomes and their role in various pathologies such as cancers and age-associated diseases such as neurological diseases [30,31]. In addition, the ability of exosomes to cross the blood–brain barrier (BBB) has made them a potential source of attractive biomarkers. Biomarkers that they carry as their cargo are useful to validate or identify mechanisms associated with neurological diseases [30,31].

Therefore, there is a need to discover the role of blood-derived pro-geronic exosomal factors that have the capacity to induce rapid aging or can be indicative of neurological diseases. Uncovering these factors can help explain the process of aging and its dynamics, thereby suggesting rational avenues for rejuvenation, neurogenesis, and earlier diagnosis of various age-associated pathologies. In addition, there have been significant efforts in developing microfluidic devices for studying exosomal cancer markers for the early diagnosis of cancer with minute starting samples. Within these devices, general exosome markers are used to verify the presence of exosomes, then cancer-specific markers are used to determine if the exosomes are derived from cancer cells and for clinical utility in cancer diagnosis. Some of these general exosome markers consist of but are not limited to, anti-CD9 [32,33,34], anti-CD63 [32,33,34,35], TSG101 [35], and CD81 [34]. In this section, we will review different devices developed for the purpose of detecting cancer and age-specific exosomes derived from patients’ serum samples.

### 4.1. Technologies for Profiling of Antigen-Specific Exosomal Biomarkers

#### 4.1.1. Immunoassay-Based Technologies

Sina et al. 2016 report the use of a surface plasmon resonance (SPR) platform for the detection of exosomes purified from breast cancer cell media and patients’ serum. The surface of the SPR chip, made from gold, is functionalized with biotinylated anti-CD9 or CD63-binding of biomolecules results in a mass increase at the sensing surface of the chip, which changes the local refractive index [32]. The cancerous exosomes were detected using a secondary human epidermal growth factor receptor 2 (HER2) antibody, and the resulting SPR signals demonstrate overexpression of HER2 biomarker on exosome samples, consistent with the previous studies [36]. This study showcases the potential clinical utility of this device for cancer exosome screening. SPR was also utilized in other studies for the detection of exosomes based on other cancer-associated exosomal surface biomarkers such as epidermal growth factor receptor (EGFR) and programmed death-ligand 1 (PDL-1) [37].

#### 4.1.2. Fluorescence and Field-Based Technologies

Surface markers are not the only important biomarkers that exosomes hold. Many age-related or cancer-related diseases receive information from the mRNA content of exosomes, which yields a need for more effective screening of exosomal mRNA cargo. Ibsen et al. 2017 repurpose the alternating current electrokinetic (ACE) microarray chip device to isolate glioblastoma exosomes from undiluted plasma. This device uses a dielectrophoretic (DEP) separation force generated by an alternating current electric field [35]. The exosomes are isolated due to their attraction to the DEP high-field regions and verified on-chip with immunofluorescence, while larger particles and cells are pulled toward the DEP low-field regions (Figure 6A). There is further exosome enrichment through CD63 antibody-coated circular microelectrodes to ensure no small particle contamination (Figure 6B) [35]. Downstream RT–PCR was done to identify glioblastoma-specific mutated EGFRvIII mRNA. The mRNA assays validated that the ACE device was able to both isolate exosomes and their mRNA [35].

Integration of multiple screening techniques allows researchers and clinicians to receive the most information available from each biofluid. Ibsen et al. 2017 integrated multiple techniques for glioblastoma exosome profiling. Additionally, Ko et al. 2017 developed a device termed, ExoTENPO to isolate specific subtypes of exosomes from plasma by magnetically labeling them and employing machine learning to determine cancer and pre-cancer individuals and mice from healthy cohorts based on exosomal mRNA cargo [33]. To do this, ExoTENPO is designed to have pores that are 600 nm in diameter to allow aggregates and cell debris to pass through. A second polycarbonate membrane with a 200 nm pore is coated with a layer of Ni_80_ Fe_20_ (magnetic material) at the edge of each pore to magnetically isolate exosomes (Figure 7Ai). Each magnetically captured exosome is labeled in two-steps with biotinylated antibodies (CD9 and CD63 for exosomes and EPCAM for cancerous subpopulations) and streptavidin-coated 50 nm magnetic nanoparticles (Figure 7Aii) [33]. This two-step process allows for 10× less antibody usage and more particle loading. Only particles bound to the magnetic nanoparticles will go toward the NiFe magnet creating a high flow, minimizing background noise from other particles. Furthermore, these magnetic approaches are favorable over optical approaches when working at nanoscales because magnetostatics is not confined to any geometry, whereas the size of optical trapping is limited by the wavelength of light. Finally, a neodymium (NdFEB) disc magnetizes both the ExoTENPO and the magnetic particles for on-chip lysing (Figure 7B). Exosomal mRNA is then isolated and profiled using qPCR, and a machine-learning algorithm is applied to predict the state of the patient.

Ramshani et al. 2019 took a similar approach to Ko in combining mechanical exosome lysing and membrane sensor technologies into an amplification-free, single-step quantification of exosome micro RNAs (miRNAs) from untreated plasma samples. Extracellular vesicles (EV) containing microRNAs (miRNAs) have tremendous potential as biomarkers for the early detection of disease. The microfluidics platform is capable of absolute quantification (<10% uncertainty) of both free-floating miRNAs and exosomal-miRNAs in plasma with 1 pM detection sensitivity [38]. The assay time is only 30 minutes as opposed to 13 hours with conventional techniques (Appendix A) [7] and requires only ~20 μL of the sample as opposed to 1 mL for conventional RT–qPCR techniques. Unlike conventional RT–qPCR methods, this technology does not require a separate step for exosome extraction, RNA purification, reverse transcription, or amplification. This platform can be easily extended for other RNA and DNA targets of interest, thus providing a viable screening tool for early disease diagnosis, prognosis, and monitoring of therapeutic response. As shown in Figure 8, the device consists of a connected two-chip system. The first chip is made of a polycarbonate microfluidic channel, which the raw plasma sample passes through, laid atop a piezoelectric substrate (Figure 8i). To the side of the microchannel, a surface acoustic wave (SAW) electrode projects along the piezoelectric surface [39]. The SAWs provide the mechanical lysing of passing exosomes (Figure 8ii) [38,40]. A connecting tube transfers the lysed sample into a second chip where the concentration and sensing of the miRNAs occur (Figure 8iii,iv). The second chip utilizes two sets of ion exchange membranes (IEMs) to concentrate and quantify the target miRNAs [38]. For IEM sensing, the surface is functionalized with oligo probes that are complementary to the target miRNAs. After hybridization, the resulting voltage shift can be directly correlated with miRNA concentration.

Measured levels of miR-21, a biomarker for liver cancer, from plasma in both mouse models and clinically relevant patients were compared against conventional methods (RT–qPCR). The device produced similar quantification results but needed lower sample volumes (20 µL versus 1 mL) and exhibited faster turnaround (20 min versus 13 h) [38,39].

The microfluidic techniques featured above represent only a sample of the ways that exosome isolation and detection can be optimized by transitioning to micro or nanoscale. Microfluidic platforms accelerate exosome biomarker discovery by conserving precious sample volumes while integrating downstream analysis of higher sensitivity than conventional methods. A comparison of all the microfluidic platforms discussed thus far is represented in Appendix A.

### 4.2. Microfluidic Approaches for Screening Neurotoxic Biomarkers

Unlike the microfluidic approaches for isolating and detecting exosomes for cancer profiling as described above, existing microfluidic techniques associated with neurodegenerative biomarkers have barely entered the space of exosomal profiling. They instead focus on the detection of soluble proteins such as tau protein and amyloid-beta in the diagnosis of Alzheimer’s disease (AD) and alpha-synuclein for Parkinson’s disease (PD). In this section, we will feature some of the microfluidic platforms used for detecting soluble proteins indicative of neurodegeneration and how they can be applied to opportunities for microfluidic screening of exosomes.

#### 4.2.1. Microfluidic Detection of Alzheimer’s Disease Biomarkers: Tau Protein and Amyloid-Beta

Costa-Rama et al. 2014 developed an electrochemical immunosensor for the detection of amyloid beta1-42 (Aβ1-42) purified protein. Amyloid-beta (Aβ) is a protein in which its accumulation within the brain presents neuropathological effects. The immunosensor consisted of screen-printed carbon electrodes with streptavidin-modified gold nanoparticles for transducing the electrochemical binding of Aβ1-42 (Figure 9Ai). The binding of Aβ1-42 to the sensing surface is facilitated with biotin-Aβ1-42 and anti-Aβ1-42 causing an analytical signal to be produced from anodic stripping of the silver electrodes by cyclic voltammetry (Figure 9Aii). The developed sensor was performed with an LOD of 0.1 ng/mL and a wide linear range between 0.5 and 500 ng/mL. This lower limit of the linear range is acceptable, given the current standard of Aβ to diagnose patients with dementia, and AD has a cutoff value around 500 pg/mL [41]. Tao et al. 2015 adopt a miniature quartz crystal microbalance (MQCM) for the detection of the same Aβ1-42 biomarker. This technique is favorable because it presents a label-free strategy for detecting aging biomarkers. The principles behind MQCM are that once the mass of a sample is loading on the piezoelectric active crystal area, there is a shift in the resonance frequency, which is directly detected by a gold-plated sensor [42]. Specific detection of Aβ42 was achieved with a cross-linker and Aβ42 antibody immobilization on the piezoelectric detecting reservoir. On-chip determination of Aβ42 from human serum was realized by correlating resonance frequency over time with the concentration of Aβ42, resulting in a linear range from 0.1 µM to 3.2 µM.

Yoo et al. 2017 utilized interdigitated microelectrode sensors (IME) and medium changes from plasma to buffer for detection of Aβ and diagnosis of AD. The IME sensors were based on an impedimetric sensor platform, which facilitated signal cancellation and amplification processing (SCAP) once a medium change occurred. An Aβ antibody-coated microchannel chip was integrated with the IME chip for sample and buffer introduction to allow for increased surface interaction with Aβ. As seen in Figure 9Bi, real-time measurement of impedance changes between the two conditions of detecting synthetic Aβ with and without the SCAP system show that integration of the SCAP system allows for signal amplification of impedance changes, whereas a system without SCAP yielded sensitivities insufficient for discrimination of Aβ. Additionally, the IME chip with the SCAP signal processing system and the plasma-buffer exchange allowed for the filtering of noise from other plasma factors and stabilization of signal for comparison of AD mice to healthy mice plasma. Figure 9Bii displays the importance of the PBS buffer exchange when using biological plasma samples. Raw noise from Aβ binding from AD mouse plasma compared to healthy mouse plasma was cleaned upon PBS injection and flushing of other plasma factors, resulting in plasma from AD mice having the larger of the two signals (Figure 9Bii) [43].

Another prominent AD biomarker is tau protein. Tau proteins, commonly found in neurons, are crucial in regulating microtubule polymerization and stabilization in the brain [44]. Patients with Alzheimer’s disease tend to have hyperphosphorylated forms of the protein tau, which are less likely to bind microtubules leading to deposition into neurofibrillary tangles (NFTs) within neurons [44,45]. Thus, it additionally has been viewed as a valuable biomarker for microfluidic aging studies such as Vestergaard et al. 2008, who developed the first biosensor for detection of tau protein through an immunoaffinity LSPR method [46]. Tau antibody was immobilized onto a self-assembled monolayer of Au nanoparticles onto a glass surface, which allows for a measurable increase in absorbance of light once there is a binding event from tau protein; all detectable with a standard UV-vis spectrometer. The LSPR, immune-based chip was able to detect tau protein to a lower limit of 10 pg/mL, suggesting it would be useful for clinical detection since the cutoff value of tau protein in the cerebral spinal fluid (CSF) for AD patient is 195 pg/mL [46].

Li et al. 2018 utilized piezoelectric sensing for the detection of tau protein in the buffer and in artificial CSF. Interestingly, they utilized a different method of secondary recognition for tau-binding outside of the conventional antibody–antigen binding. Li investigated if tubulin could be used as an alternative secondary recognition element for tau protein biosensing [47]. It is well known that tau is a microtubule-associated protein and interacts with tubulin, specifically in microtubule polymerization and stabilization [48]. Piezoelectric sensing is useful in this application in that any kinetically favorable interaction of tau with tubulin will result in mass loading detectable by the quartz crystal alternating current. In comparison to a traditional tau-antibody-based sandwich assay and the tau-tubulin assay, the novel tau-tubulin secondary interaction shows an extended range of dose-dependent frequency shifts with tau-tubulin. Here, the biochemical reaction of tau-tubulin binding may be measured in real-time without the use of any labels and without the risk of cross-reactivity as seen with antibody-based methods. This method provides some insight on alternative methods for surface functionalization outside of immune-based antigen–antibody binding and shows the potential use of kinetic interactions of biomarkers for detection.

The neurodegenerative biomarkers discussed thus far have focused solely on Alzheimer’s disease. However, other neurological diseases such as Parkinson’s disease and dementia with Lewy bodies (DLB) are strongly correlated with another neurotoxic biomarker, namely alpha-synuclein (aSyn). It has been suggested that, like other neurotoxic biomarkers, aSyn can form aggregated oligomers that may result in various synucleinopathies [49]. Horrocks et al. 2015 aimed to detect kinetic changes between amorphous oligomers, which have low cell cytotoxicity and structural rearrangement to form the proteinase-K resistant and heavily cytotoxic oligomers using single-molecule Forster resonance energy transfer (FRET). Using a microfluidic chip under “fast-flow,” essentially with a velocity of 2 cm/s, a diluted sample of fluorescently labeled aSyn oligomers and monomers were injected into the sample inlet, with the outlet connected to a pump for inducing a negative pressure and the fast-flow. The device itself is mounted onto a confocal microscope in which 488 nm radiation is focused down, and fluorescence from 488 nm and 594 nm (excited indirectly from the dichroic mirror) is detected by avalanche photodiode detectors (APDs) (Figure 10Ai). Cytotoxic oligomers have a more complex, aggregated conformation and a larger FRET signal compared to less cytotoxic oligomers and monomers (Figure 10Aii) [50].

In a different application of microfluidics, Fernandes et al. 2016 utilized microfluidic platforms in a cell culture setting to study the paracrine communication of reactive oxygen species and alpha-synuclein between cellular microenvironments of neuroglioma and microglia cells for studying mechanisms associated with PD. In this study, the microfluidic device is built around two cell culture chambers, which are connected by three channels (Figure 10Bi) [51]. This device uniquely integrates pneumatic valves that enable isolation and controlled communication of cell populations through soluble molecules. Furthermore, cellular communication is achieved through diffusion or by perfusion based on the distance between the two chambers is 250 µm. This minute distance allows for released molecules to quickly diffuse from one cell population to another. Pneumatic valves further control perfusion-based communication of ROS and aSyn bidirectionally from one cell population to the other, as seen in Figure 10Bii. As shown in Figure 10Bii the blue path and white arrows indicate potential ROS and aSyn communication, and the red bars function as controllable pneumatic valves. The control of fluids on a micro-scale aid in monitoring and controlling of diffusion and convection effects of the cellular milieu and the chemical stimuli produced by cell populations, which is much more difficult in the mL scale in a traditional culture plate model [51].

#### 4.2.2. Opportunities to Develop Technologies for Profiling of Exosomal Cargo Biomarkers

Although all the microfluidic techniques discussed with neurodegeneration thus far have not involved exosomes, they have involved the most used biomarkers for the diagnosis of neurodegenerative diseases. Despite the promise of microfluidic platforms for protein detection, they still suffer from low protein concentration within the blood, given that soluble proteins are metabolized quickly within the circulation. Interestingly, emerging studies have shown that neural exosomes carry some of the same neurotoxic proteins’ indicative of neurodegenerative diseases such as tau protein, alpha-synuclein, and amyloid-beta (Figure 11B) as their cargo and pass the blood–brain barrier (BBB) into the normal blood circulation (Figure 11A). As exosomal cargo, these biomarkers are protected from degradation as an exosome flows through circulation [52,53,54]. Furthermore, neurotoxic proteins have been found to be associated with exosomes isolated from AD and PD patients’ brains, CSF, and blood [55,56,57,58]. However, no functional assays have truly determined whether exosomal tau, alpha-synuclein, or amyloid-beta contributes to pathology.

Studies such as Asai et al. 2015 incubated microglia with pre-aggregated human tau (hTau) protein to detect the presence of microglia phagocytose and tau protein in exosomes. Asai et al. demonstrated that exosomes isolated from mouse brains of tauopathic and AD mice contain hTau oligomers and that microglial depletion reduces hTau content in exosomes [57]. Polanco et al. 2016 proposed the notion that exosome-like extracellular vesicles carry tau seeds that could contribute to tau pathology in Alzheimer patients. Upon exosome extraction from mice brains with hereditary tauopathy, ten times more tau protein was found in the exosomes from the tauopathic mice compared to the wild-type (WT) mice, in a ratio of 1:12.5 (WT: tauopathic) [58]. Sinha et al. 2018 highlighted that exosomes isolated from the brain of deceased AD patients contain increased levels of amyloid-beta oligomers (oAβ), confirmed with exosomal and oAβ immunofluorescence co-localization. Additionally, the exosomes derived from AD patient brains were assessed for their ability to transport neurotoxic oAβ to nearby healthy neurons within a cell culture model. The results from these studies suggested that oAβ were indeed found within exosomes and in the previously heathy neuroblastoma cells which now had morphology indicative of neurodegeneration [59]. Exosomes have also been implicated in other neurodegenerative disorders such as DLB and Parkinson’s disease. Ngolab et al. 2017 sought to determine if exosomes carrying the aSyn protein cargo specific for DLB could transmit pathology. DLB progression is characterized by clusters of aSyn within Lewy Bodies and neuron cell terminals [60]. Alpha-synuclein is a protein typically found within the presynaptic ends of neurons and is suggested to have a role in synaptic vesicle formation and mobility [61]. To assess if exosomes derived from the DLB brain could transmit DLB pathology to non-diseased tissue, exosomes isolated from DLB brain tissue were confirmed for the presence of aSyn and injected into the cerebrum of non-diseased mice. Results displayed a higher level of phosphorylated aSyn in the mice that received DLB exosomes compared to healthy controls. Additionally, mice that were injected with DLB exosomes showed a higher expression of aSyn in their neurons, suggesting that exosomes are internalized by neurons [60].

Many neurodegenerative disorders are diagnosed later in life and mainly through cognitive and behavioral assessments, which can be subjective. Thus, there is an unmet need for a more objective means of earlier diagnosis of degeneration. The studies presented above implicate the role of exosomes in carrying neurotoxic proteins while also discovering some of the mechanisms to which they propagate these proteins to healthy cells. Although these studies serve as a great benchmark for biomarker discovery, there is still a considerable unknown associated with the numerous pathways involved in aging and neurodegeneration. Therefore, technologies to examine the biomarkers (within exosomes) can yield information predictive of the onset of neurological disorders. The microfluidic screening tools discussed in the previous sections can assist in profiling exosomal cargo from neural cells and can be applied to many degenerative disorders in hopes of advancing the therapeutic outcomes of these disorders. Microfluidic exosome capture and lysing techniques featured in Ko 2017 and Ramshani 2019 could be utilized for isolation of neural exosomes from blood and freeing of their neuro biomarker cargo. Furthermore, once freely soluble, these neuro biomarkers can be detected for their clinical relevance and AD or PD diagnosis in a similar fashion to the previously described methods [41,42,43,47,49,50,51]. Any combination of these techniques can be applied to enhance microfluidic neurodegenerative exosome research and offer a substantial effort in furthering more facile, real time, early detection of neurodegenerative disorders.

## 5. Challenges to Commercialization

Many of the techniques featured within this review exploit the antibody/antigen relationship between surface markers of exosomes to offer increased specificity and enhanced interaction time with the biosensing surface. However, the use of antibodies in biomarker profiling can hinder a technology from advancing to a point-of-care setting. For example, based on the World Health Organization’s (WHO) criteria for commercializing a technology to the point-of-care setting, REASSURED (Table 1) [62], immunoassays are always at risk for false positives if the antibody of choice has any cross-reactivity with any biological targets. Furthermore, to accommodate the rapid and robust requirement, some antibody incubation steps still require a few hours within a microfluidic device and may require refrigeration. It may serve well for microfluidic devices to move from profiling not only surface markers of exosomes but also cargo to accommodate these needs for commercialization.

Additionally, although many of the studies in this review show correlation of exosome properties with disease states such as concentration or cargo, it does not provide a clear mechanism of exosome influence on disease states to inform on clinical decision-making. The incorporation of machine learning techniques downstream of microfluidic exosome isolation and profiling will remedy this obstacle. As seen with Ibsen et al. 2017, machine learning can be incorporated to profile exosome biomarkers and build a training set to be used as a benchmark for unknown clinical samples to make more informed decisions. The caveat with machine learning models, however, is the need for very large data sets to accommodate clinical utility in humans. Training data sets for machine learning techniques sometimes need hundreds of samples to accurately predict the influence a biomarker has on clinical outcomes [63,64,65]. Although this large sample requirement may be daunting, it appears that many diagnostic fields are moving towards incorporating machine learning into their downstream analysis to help provide more information to clinicians.

## 6. Conclusions

The role of exosomes in diagnostics and therapeutics has yet to reach its potential. Any obstacles to progress in its field remain with the fact that traditional isolation and detection methods are cumbersome, require intensive equipment and workflow, and lack of standardization for the validation of exosomes once isolated. Fortunately, most of the microfluidic tools covered in this review address some of these challenges and provide an avenue for the ongoing optimization of microfluidic exosome isolation and detection techniques. These microfluidic techniques enable the development of the field through the ease of use, decrease in isolation and detection time and complexity.

Furthermore, exosomes can provide a non-invasive alternative for liquid biopsies for difficult disorders such as those of cancers and neurodegenerative diseases. However, for these fields to evolve further, there should be a coordinated advancement in the traditional and microfluidic techniques that stand as the gatekeepers to exosome research. Microfluidic profiling of the exosomes released from cancer cells can provide a highly sensitive, less invasive outlook on the heterogeneity of cancers for designing more optimal diagnostics and therapeutics. Similarly, profiling exosome cargo released from neural and other germ cell tissues may present a channel to decipher the mechanisms behind age-related diseases. In conclusion, as standardization and microfluidic techniques improve the field, exosomes will continue to enhance precision-medicine both in the fields of age-related cancers and neurodegenerative diseases.

## Figures and Tables

**Figure 1 molecules-26-00535-f001:**
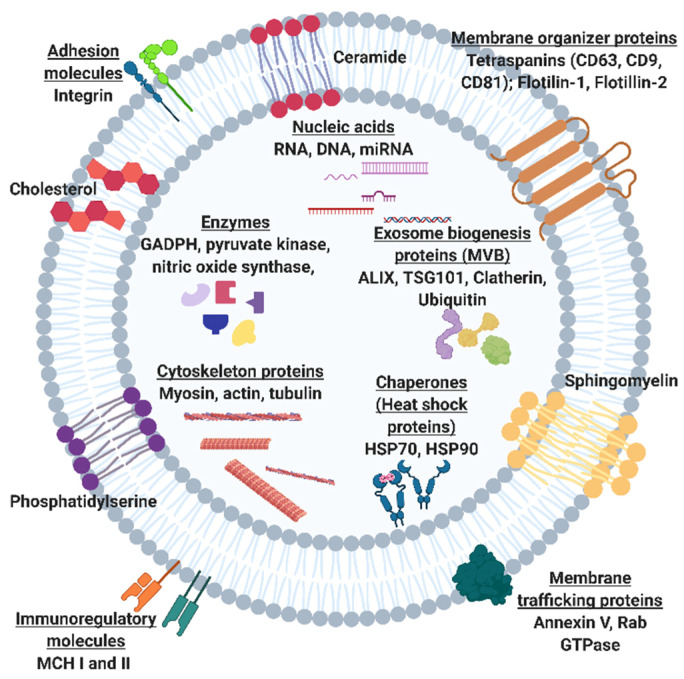
Schematic of exosome including intracellular cargo and surface proteins and macromolecules. Common exosome surface markers include tetraspanins CD63, CD81, CD9, flotillins 1 and 2. Common intracellular markers include exosome biogenesis markers ALIX, a programmed cell death 6 interacting protein, and tumor suppressor gene 101 (TSG101) and enzymes such as glycolytic enzyme, Glyceraldehyde 3-phosphate dehydrogenase (GADPH) and nitric oxide synthase.

**Figure 2 molecules-26-00535-f002:**
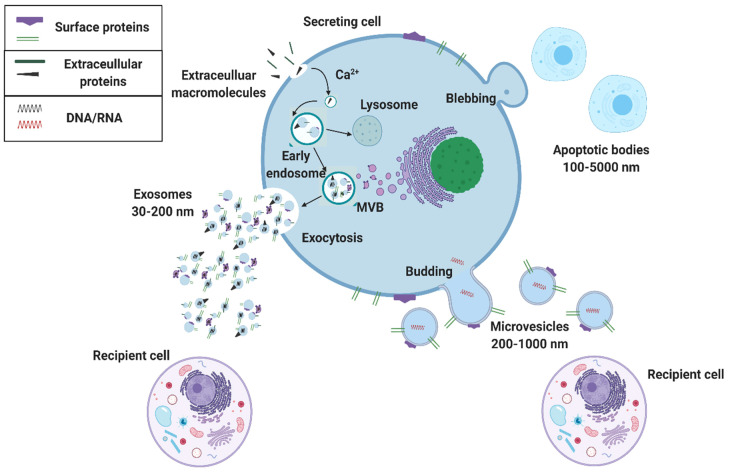
Extracellular vesicle biogenesis. Extracellular vesicles that are secreted from a cell include apoptotic bodies (100–5000 nm), microvesicles (200–1000 nm), and exosomes (30–200 nm). Exosomes are made from an early endosome pathway and can contain information about both the parent cell’s organelles and plasma membrane, whereas microvesicles merely bud and contain membrane information from the parent cell’s surface membrane.

**Figure 3 molecules-26-00535-f003:**
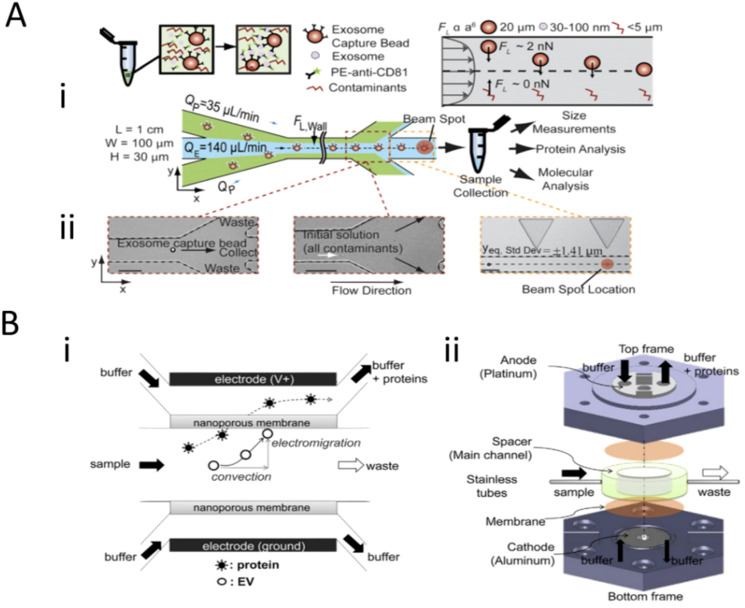
Size-based and combination techniques for the isolation of exosomes. (**A**) Workflow for the rapid inertial solution exchange for the enrichment and flow cytometric detection of microvesicles using exosome capture beads and microfluidic flow system. (**B**) Schematic of extracellular vesicle isolation with electrophoretic migration techniques using a nanoporous membrane and Faradaic reactions between device electrodes. (**A**) and (**B**) parts of figure were adapted with permission under the Copyright Clearance Center.

**Figure 4 molecules-26-00535-f004:**
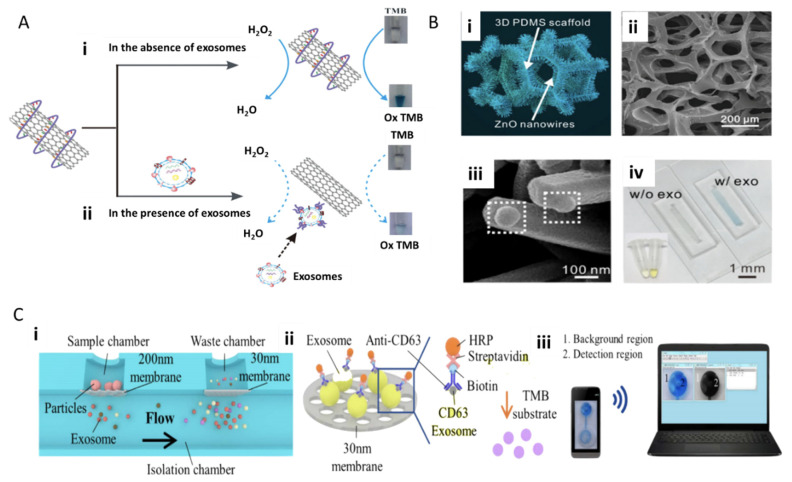
Colorimetric, surface-functionalized biosensing of exosomes. (**A**) Aptamer-based biosensor utilizing single-walled carbon nanotubes (s-SWCNTs) for immunocapture of exosomes with CD63 and subsequent colorimetric 3,3′,5,5′-tetramethylbenzidine (TMB) assay. (**Bi**) ZnO-based biosensor applying ZnO nanowires, polydimethylsiloxane (PDMS) scaffold structure with ZnO nanocrystals forming nanowires. (**Bii**) SEM of ZnO nanowires on the scaffold. (**Biii**) SEM image of exosome capture. (**Biv**) Colorimetric assay example with and without exosomes with sandwiched immunocapture of CD9 and CD63 functionalized ZnO nanowires (reprinted with permission from Copyright Clearance (2016) and (2017) Elsevier). (**Ci**) Schematic of a double-filtration microfluidic device for size-based exosome isolation featuring the initial 200 nm membrane and subsequent 30 nm membrane. (**Cii**) Schematic of direct enzyme-linked immunosorbent assay (ELISA) for extracellular vesicle (EV) detection with on-chip ELISA. (**Ciii**) Detection of colorimetric HRP-TMB interaction with smartphone and computer. Reprinted with permission from Copyright Clearance (2017) Nature Scientific Reports.

**Figure 5 molecules-26-00535-f005:**
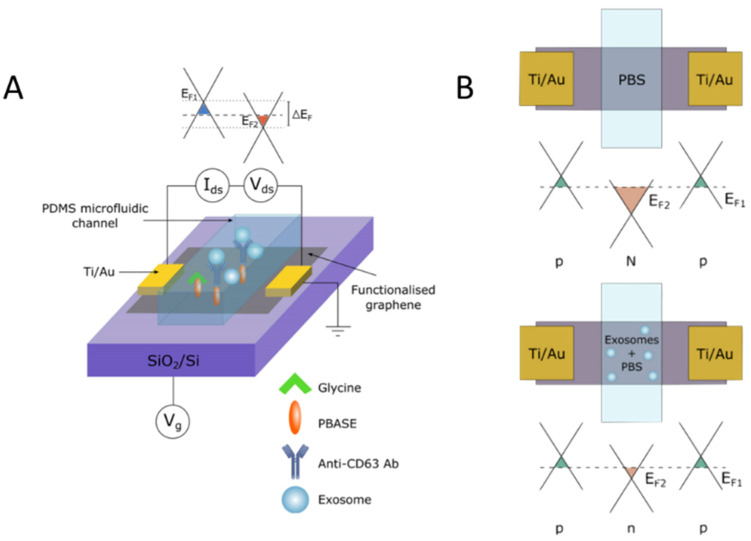
Antibody surface functionalization for exosome capture and electric biosensing with graphene. (**A**) Schematic of gFET mechanism for detection of negatively charged exosomes once captured by the graphene surface through surface functionalization with a chemical linker and anti-CD63 antibody. (**B**) Schematic of negative charge from exosome surface potential leading to positive charge accumulation in the graphene surface, showing Fermi energy of graphene surface. Reprinted with permission from Copyright Clearance (2019) Springer Nature Scientific Reports.

**Figure 6 molecules-26-00535-f006:**
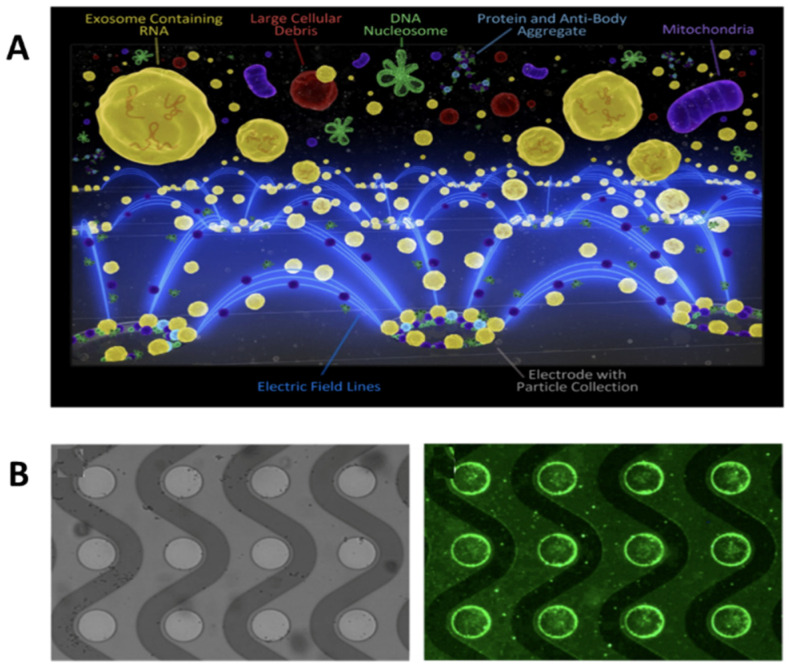
Schematic of exosomes and other particles separated with the alternating current electrokinetic (ACE) microarray chip. (**A**) ACE microarray chip fluorescent schematic of dielectrophoretic separation force pulling exosomes to microelectrodes. (**B**) Bright-field image of the circular microelectrodes (left) and fluorescent image of CD63 at the electrode edges (right) (reprinted with permission from Ibsen et al. American Chemical Society 2017).

**Figure 7 molecules-26-00535-f007:**
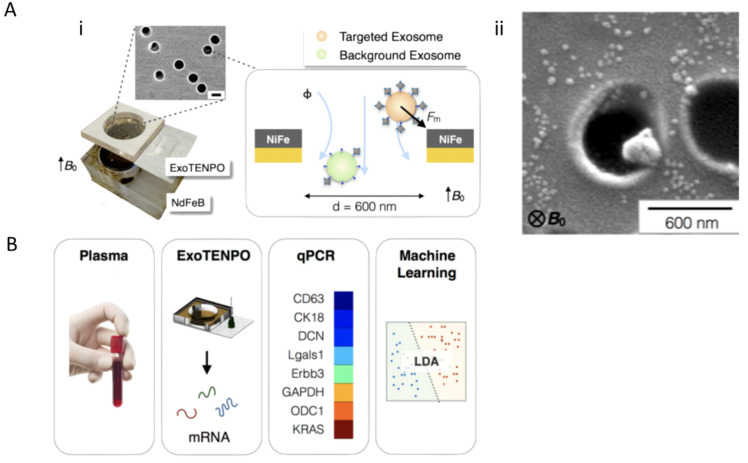
Example of combining exosome profiling with machine learning techniques for cancer diagnostics. (**A**) Schematic and workflow of exosome (**Aii**) tracked-etched magnetic nanopore chip (ExoTENPO) and the (**Ai**) NdFeB external magnet for detection of pancreatic cancer exosomes from plasma through (**B**) qPCR and machine-learning algorithms (Adapted with permission from Ko. J et al. 2017 American Chemical Society Nano).

**Figure 8 molecules-26-00535-f008:**
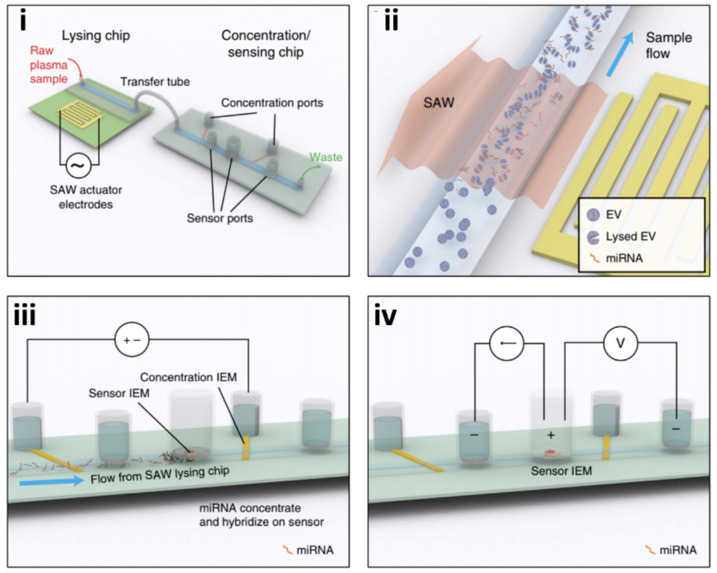
Example of the on-chip lysing method for the detection and profiling of exosome content. Schematic of the integrated microfluidic platform that features two chips, (**i**,**ii**) one for exosome mechanical lysing with surface acoustic waves (SAW) and (**iii**,**iv**) enrichment followed by a sensing chip for exosome contents for disease profiling (reprinted with permission from Ramshani et al. 2019 under Springer Nature from the Copyright Clearance Center).

**Figure 9 molecules-26-00535-f009:**
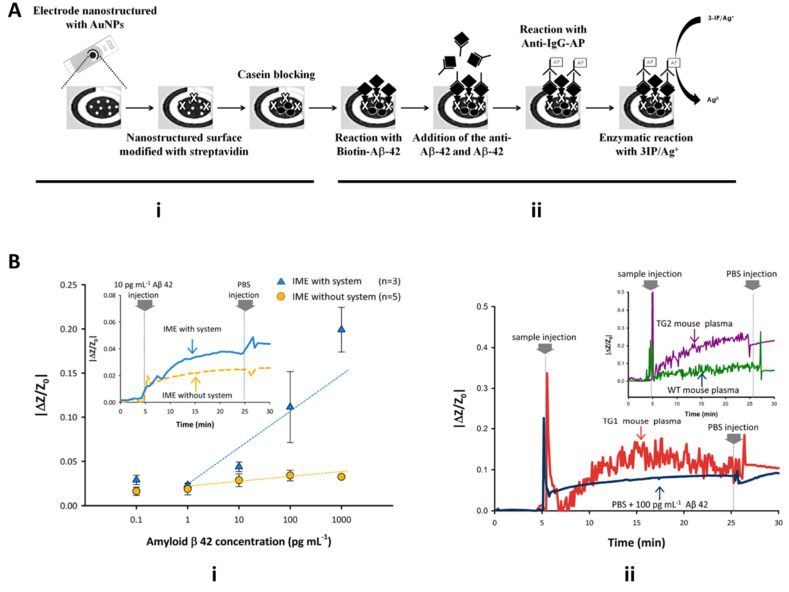
Microfluidic devices for detection of Alzheimer’s disease biomarkers. (**A**) Schematic of Costa Rama electrochemical immunosensor for detection of amyloid-beta. Sensing electrode features streptavidin-modified gold nanoparticles increase exosomal surface interaction for detection with anti-Aβ and cyclic voltammetry (adapted with permission from Costa Rama et al. 2014 and provided by Elsevier and Copyright Clearance Center. (**B**) Yoo 2017 interdigitated microelectrode sensor (IME) and impedimetric sensor readout for detection Aβ with and without SCAP system and PBS buffer exchange (reprinted under open access within Creative Commons CC BY license, Springer Nature).

**Figure 10 molecules-26-00535-f010:**
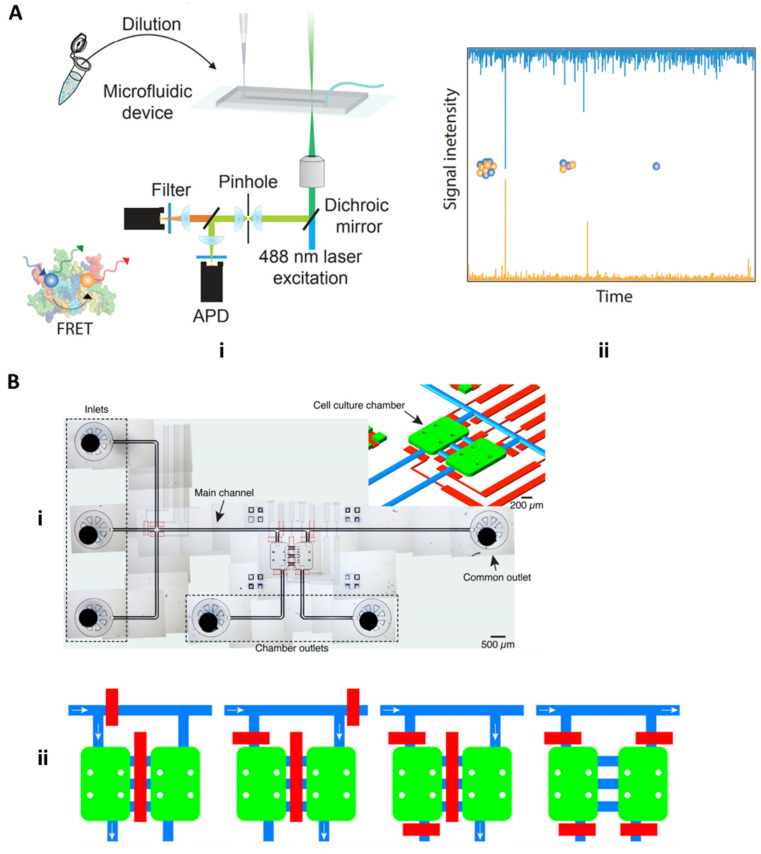
Parkinson’s disease microfluidic devices. (**A**) Horrocks 2015 microfluidic device for detecting amorphous, less cytotoxic aSyn monomers and oligomers and highly cytotoxic oligomers with FRET (adapted with permission from Horrocks et al. 2015 under copyright 2015 American Chemical Society). (**B**) Fernandes 2016 cell culture microfluidic device with pneumatic channel pumps for bidirectional controlled flow of cellular milieu and cell–cell communicative factors (adapted under open access within Creative Commons CC BY license, Frontiers in Neuroscience).

**Figure 11 molecules-26-00535-f011:**
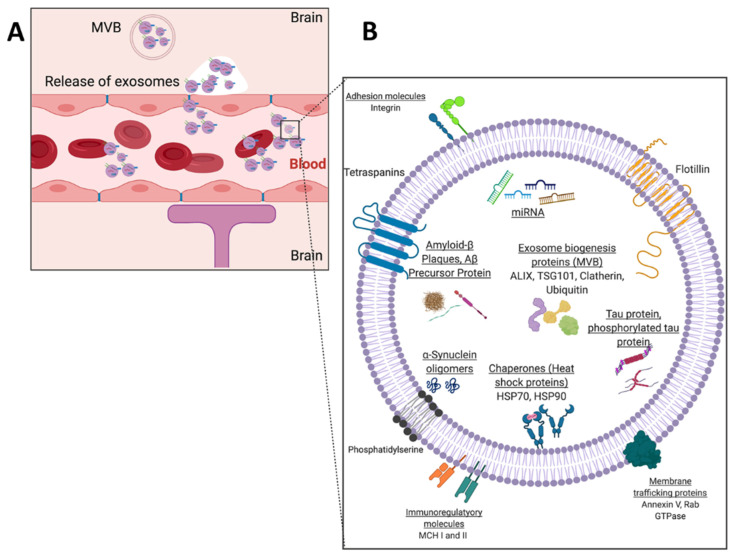
(**A**) Blood–brain barrier schematic with exocytosis of neural-derived exosomes from brain cells into bloodstream and (**B**) schematic example of a neural exosome.

**Table 1 molecules-26-00535-t001:** REASSURED is the acronym for the criteria under the World Health Organization’s (WHO) current requirements for commercialization of technology to a point-of-care setting.

	Criteria	Description
R	Real-time connectivity	Tests are connected, and/or a reader or mobile phone is used to power the reaction and/or read the test results to give appropriate data to decision-makers
E	Ease of specimen collection	Tests should be designed for use with non-invasive specimens
A	Affordable	Tests are affordable to end-users and health systems
S	Sensitive	Avoid false-negatives
S	Specific	Avoid false-positives
U	User-friendly	The procedure of testing is simple with few steps and little training
R	Rapid and robust	Results are available for giving treatment within the first visit (15 min to 2 h); Tests can survive as stock without additional transport or storage like refrigeration
E	Equipment-free or simple environment	The test does not require any special equipment
D	Deliverable to end-users	Accessible to those who need the tests

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
