# Peer review of "The Microfluidic Toolbox for Analyzing Exosome Biomarkers of Aging"

_molecules, 2021, doi:10.3390/molecules26030535_

Round 1

Reviewer 1 Report

The manuscript by DeCastro et al. looked at various techniques for high-throughput isolation and detection of exosomes using microfluidic and biosensing strategies. They also discussed exosome-based researches which explored age-related diseases including cancers and neurodegenerative disorders. I have a few concerns regarding this paper which are described in the following.

This review article is very long (~12,000 words, >2000 words only for introduction) and discuss each paper with too much unnecessary details which make it hard to read. I was not comfortable following the flow of the paper and it seems more like a series of repot paragraphs summarizing the whole text of each research paper. I believe that this paper needs to be significantly edited to make it more readable. To acquire the qualities of a good review paper, authors should bring the word counts to less than 6,000 and only mention important and useful information for readers. One of the main purposes of review paper is to motivate readers to take a closer look at the discussed research when they got a general idea about what interests them. I brought a few of obvious inconsistencies I detected in the text.

In line 598, author started a paragraph with “In previous work of Asai et al.” while there was no mention of any works by Asai et al. before.

In Line 612, authors talk about protein tau while they have proper introduction to tau protein in line 617.

In line 557, authors mention the research by Zeng et al. which they have improved the plasmonic interferometer array. However, there is no discussion of this method before this.

In line 641, what is DLB?

In line 123, mm should be nm.

Author Response

REVIEWER 1

The manuscript by DeCastro et al. looked at various techniques for high-throughput isolation and detection of exosomes using microfluidic and biosensing strategies. They also discussed exosome-based researches which explored age-related diseases including cancers and neurodegenerative disorders. I have a few concerns regarding this paper which are described in the following.

 In this revision, as per all the reviewer suggestions, we have altered the manuscript to focus on the main thesis of high throughput screening mechanisms for exosomes and applications for exosome screening and sensing for aging. In following this thesis, we have a more focused view on a range of microfluidic technologies for exosome isolation and detection which has subsequently condensed the manuscript to within 6700 words. Furthermore, to balance the types of technologies discussed, new microfluidic platforms were introduced with detailed descriptions of device design with advantages and disadvantages of those designs. After careful consideration and to accommodate this more diversified profile of techniques, some of the previous techniques that received reviewer comments have been removed. We believe that in doing so, this has improved the manuscript and has addressed reviewer concerns. We thoroughly inspected all figures to have detailed in-text descriptions and citations to be consistent throughout the manuscript. We additionally incorporated a new section for microfluidic antigen detection of neurodegenerative biomarkers and our future perspective of how they can be applied to exosome research. Finally, we discussed improvements required for these devices to follow through with future commercialization. We thank the reviewers for their thoughtful comments and questions about the structure, organization, balance, and device details of our manuscript.  We hope that we were able to address their questions and concerns with the modifications incorporated into this resubmission.

This review article is very long (~12,000 words, >2000 words only for introduction) and discuss each paper with too much unnecessary details which make it hard to read. I was not comfortable following the flow of the paper and it seems more like a series of repot paragraphs summarizing the whole text of each research paper. I believe that this paper needs to be significantly edited to make it more readable. To acquire the qualities of a good review paper, authors should bring the word counts to less than 6,000 and only mention important and useful information for readers. One of the main purposes of review paper is to motivate readers to take a closer look at the discussed research when they got a general idea about what interests them. I brought a few of obvious inconsistencies I detected in the text.

We thank the reviewer for their comments. In our previous work, our introduction and the manuscript alone were >2000 words and ~12,000 words, respectively. In this resubmission, we have re-written the introduction to highlight the main thesis of the manuscript within the first 500 words. We believe that this more concise description gives the reader a simple point of reference on the importance of the current topic which is enhancing microfluidic screening tools for exosomal aging research. Additionally, we covered many microfluidic techniques that spanned a wide range of specific details of each device. Based on the reviewer’s feedback, we have instead highlighted the important findings of the specific studies and the only included specific device details that made it unique for exosome isolation or detection. With these adjustments we have the word count around 6700 words for the full manuscript.

In line 598, author started a paragraph with “In previous work of Asai et al.” while there was no mention of any works by Asai et al. before.

We thank the reviewer for raising this point. We would like to clarify that line 598 “In previous work of Asai et al.” in fact was misleading because no previous work of Asai had been discussion. This line has now been re-written as “Studies such as Asai et al. 2015” line 625.

In Line 612, authors talk about protein tau while they have proper introduction to tau protein in line 617.

We thank the reviewer for raising this point. We would like to apologize for the inconsistencies between tau protein introduction for lines 612 and 617. To address this, we have added an introductory summary of tau protein in the resubmission within lines 508-511.

In line 557, authors mention the research by Zeng et al. which they have improved the plasmonic interferometer array. However, there is no discussion of this method before this.

We thank the reviewer for their comment. However, in addressing a comment made by Reviewer 2 on the balance of different techniques that were covered, the discussion of Zeng et al. plasmonic interferometer array was removed from the manuscript for this resubmission. Thus, no formal discussion on the principles behind plasmonic interferometer arrays were included in the text of this resubmission.

In line 641, what is DLB?

We thank the reviewer for raising this point. We would like to apologize for not reporting appropriate definitions of acronyms when first introduced in the text. In the previous work, line 641 where DLB was first introduced, is now defined as Dementia with Lewy Bodies (DLB) line 537-538 in this resubmission.

In line 123, mm should be nm.

We thank the reviewer for raising this point. We would like to apologize for the misrepresentation of the units within our description of Zhang et al. subpopulation of exosomes. In our previous work, line 123 which stated “exosomes (90-120 mm), small exosomes (60-80 mm), and ~35 nm diameter nanoparticles that they call ‘exomeres” has now been relabeled in this resubmission as line 162 exosomes (90-120 nm), small exosomes (60-80 nm)”.

We thank the reviewers for all their thoughtful comments and concerns. We hope we were able to address them appropriately in this resubmission and believe that with their comment, this resubmission has been greatly improved.

Reviewer 2 Report

DeCastro and co-workers reported a review manuscript regarding the microfluidic biosensors, detection, and profiling of exosome for antiaging and neurological applications. I feel the manuscript is difficult to read-only information are informed, but no discussion provided. In particular, the induction part is not convinced; several main ideas are putting in one paragraph which is not ideal. I feel it can be improved by giving less main idea and discuss why it is important to be discussed, not just only giving blend information.

For the main section, I would suggest the authors discuss more regarding the microfluidic designs and fabrication methods; the roles of obstacles and geometry in microfluidic platforms discussion could benefit readers. Further, it would be fascinating to readers if the author could discuss why and how each particular geometry was employed with pros and cons and controversy in the field and suggestion based on authors experiences or references. 

Of note, each section should be comprised of the introduction, main idea, supportive references, examples, and its own future perspective.  

I also concern regarding the balance of each topic. It seems that the authors have an obvious interest in some methodologies rather than others, which could be misleading the readers through the manuscript. 

Since the EV topic is drawing attention to many readers from widespread fields, it would be nice to see an apparent and concrete future perspective of EV. In particular, point-of-care diagnostics and commercialisation should be discussed comprehensively with ASSURE criteria (see WHO criteria for point-of-care diagnostics) and smartphone-based diagnostics.  

Thus, I feel this manuscript can be improved and is not ready to be published in this form.

Author Response

Reviewer 2

DeCastro and co-workers reported a review manuscript regarding the microfluidic biosensors, detection, and profiling of exosome for antiaging and neurological applications. I feel the manuscript is difficult to read-only information are informed, but no discussion provided. In particular, the induction part is not convinced; several main ideas are putting in one paragraph which is not ideal. I feel it can be improved by giving less main idea and discuss why it is important to be discussed, not just only giving blend information.

 In this revision, as per all the reviewer suggestions, we have altered the manuscript to focus on the main thesis of high throughput screening mechanisms for exosomes and applications for exosome screening and sensing for aging. In following this thesis, we have a more focused view on a range of microfluidic technologies for exosome isolation and detection which has subsequently condensed the manuscript to within 6700 words. Furthermore, to balance the types of technologies discussed, new microfluidic platforms were introduced with detailed descriptions of device design with advantages and disadvantages of those designs. After careful consideration and to accommodate this more diversified profile of techniques, some of the previous techniques that received reviewer comments have been removed. We believe that in doing so, this has improved the manuscript and has addressed reviewer concerns. We thoroughly inspected all figures to have detailed in-text descriptions and citations to be consistent throughout the manuscript. We additionally incorporated a new section for microfluidic antigen detection of neurodegenerative biomarkers and our future perspective of how they can be applied to exosome research. Finally, we discussed improvements required for these devices to follow through with future commercialization. We thank the reviewers for their thoughtful comments and questions about the structure, organization, balance, and device details of our manuscript.  We hope that we were able to address their questions and concerns with the modifications incorporated into this resubmission.

 We thank the reviewer for their comments. In our previous work, our introduction and the manuscript alone were >2000 words and ~12,000 words, respectively. In this resubmission, we have re-written the introduction to highlight the main thesis of ‘high throughput screening mechanisms for exosomes and applications for exosome screening and sensing for aging’ has been adapted and the content of the text has been reformatted to accommodate this more focused view. This main thesis was achieved within the first 500 words.

For the main section, I would suggest the authors discuss more regarding the microfluidic designs and fabrication methods; the roles of obstacles and geometry in microfluidic platforms discussion could benefit readers. Further, it would be fascinating to readers if the author could discuss why and how each particular geometry was employed with pros and cons and controversy in the field and suggestion based on authors experiences or references. 

Of note, each section should be comprised of the introduction, main idea, supportive references, examples, and its own future perspective.  

We thank the reviewer for their comments. We apologize for not discussing a more in-depth display of the microfluidic designs and fabrication methods. In this resubmission, we have included pros and cons of the device design and how those specific geometries aid in either exosome isolation and detection or antigen profiling. Additionally, we have provided introductions to each main section and comparisons between techniques depending on the application. After first addressing the microfluidic tools that support exosome isolation and detection, we provided a new section of microfluidic tools for specific neurological biomarkers and a future perspective on how these could be integrated into microfluidic, exosome-based neurodegenerative research.

I also concern regarding the balance of each topic. It seems that the authors have an obvious interest in some methodologies rather than others, which could be misleading the readers through the manuscript. 

We thank the reviewer for their comments. We apologize is the manuscript appeared to favor some technologies over the others. After careful consideration of the manuscript, we realized that we heavily favored size-based techniques. To remedy this for the resubmission, we only incorporated the size-based techniques that were the most impressive to us and added different technologies for exosome isolation and detection. Additionally, we incorporated examples of combined techniques and a new section on varying microfluidic techniques for detecting neurodegenerative biomarkers that may be subsequently applied to exosome research.

Since the EV topic is drawing attention to many readers from widespread fields, it would be nice to see an apparent and concrete future perspective of EV. In particular, point-of-care diagnostics and commercialisation should be discussed comprehensively with ASSURE criteria (see WHO criteria for point-of-care diagnostics) and smartphone-based diagnostics.  

We thank the reviewer for their comments and providing information on adding this interesting perspective. For this resubmission, we have reviewed the updated WHO criteria for “REASSURED” and included our future perspective of these devices and what components of the devices should be improved upon to truly follow through with point-of-care diagnostics and commercialization.

Thus, I feel this manuscript can be improved and is not ready to be published in this form.

We thank the reviewers for all their thoughtful comments and concerns. We hope we were able to address them appropriately in this resubmission and believe that with their comment, this resubmission has been greatly improved.

Reviewer 3 Report

  1. Reference format needs to be consistent.
  2. The contents lean on the device development, I will recommend that the author change its paper title to fit the content they currently have.

Author Response

Reviewer 3

In this revision, as per all the reviewer suggestions, we have altered the manuscript to focus on the main thesis of high throughput screening mechanisms for exosomes and applications for exosome screening and sensing for aging. In following this thesis, we have a more focused view on a range of microfluidic technologies for exosome isolation and detection which has subsequently condensed the manuscript to within 6700 words. Furthermore, to balance the types of technologies discussed, new microfluidic platforms were introduced with detailed descriptions of device design with advantages and disadvantages of those designs. After careful consideration and to accommodate this more diversified profile of techniques, some of the previous techniques that received reviewer comments have been removed. We believe that in doing so, this has improved the manuscript and has addressed reviewer concerns. We thoroughly inspected all figures to have detailed in-text descriptions and citations to be consistent throughout the manuscript. We additionally incorporated a new section for microfluidic antigen detection of neurodegenerative biomarkers and our future perspective of how they can be applied to exosome research. Finally, we discussed improvements required for these devices to follow through with future commercialization. We thank the reviewers for their thoughtful comments and questions about the structure, organization, balance, and device details of our manuscript.  We hope that we were able to address their questions and concerns with the modifications incorporated into this resubmission.

  1. Reference format needs to be consistent.

We thank the reviewer for their comments. We would like to apologize for the inconsistencies of the citation format, in-text and within the bibliography. We have reformatted all citations to be consistent with a subscript if in-text and brackets if within a table. Citation format in the bibliography with author names, publications, years, etc. have also been reformatted for this resubmission.

  1. The contents lean on the device development, I will recommend that the author change its paper title to fit the content they currently have.

We thank the reviewer for their comments. In our previous work, there was a heavy emphasis on device development. In this resubmission, the main thesis of ‘high throughput screening mechanisms for exosomes and applications for exosome screening and sensing for aging’ has been adapted and the content of the text has been reformatted to accommodate this more focused view. Thus, we have changed the title of the paper to “The Microfluidic Toolbox for Analyzing Exosome Biomarkers of Aging”.

We thank the reviewers for all their thoughtful comments and concerns. We hope we were able to address them appropriately in this resubmission and believe that with their comment, this resubmission has been greatly improved.

Reviewer 4 Report

 Comments on the paper “Microfluidic Biosensing, Detection, and Profiling of Exosomes for Aging and Neurological Diseases” by   Jonalyn DeCastro et al

Topic of this review is quite interesting and timely.   I am very much impressed with the depth and breadth of coverage of the bio sensing aspects and the discussion on the limits of detection in each method.  There are two important elements in this review, while one is the bioanalysis and medical related, and the other one is microfluidic platforms and exosome detection techniques, which authors balanced well in their manuscript.    However, microfluidics aspects of the review require a bit more attention and this can improve the quality and usefulness of the paper further.  Following are some of my specific comments.

  1. Not sure, who is the author from University of California, Berkeley.  Affiliation is given but no author’s name is related to this.
  2. Line 59-60, The device is comprised of a simple polydimethylsiloxane  (PDMS) microchannel …. field (Figure 3B).  Something missing here. 
  3. Line 75, “…..toward the two-sided  ”  Does this mean  to say “ towards the outlets on either side of the middle outlet?
  4. Line 169-170, “The detection aspect…detection area”  This sentence require some rewriting to make it clear.
  5. Line 178, Not sure, what figure XA referring to? 
  6. Line 234-236, Not sure what the figure 1a and figure. 1b referring to?   Are these referring to figure6a and 6b respectively?
  7. Line 248-250, The nano interfaced…detection.. This sentence is not clear…..
  8. Line 307-308, Language correction
  9. Many figure explanations  (in the text) were not relating the figure details well. For example,  8A and 8B are not explained well in the text.  With fig. 8B,  Authors should explain device 1 and device II parts and how the device design helped to separate exosomes by size. Otherwise, providing the figure doesn't help.
  10. Line 364-366, A polyethylene glycol film to promote particle scattering and directs light toward the CMOS sensor, allowing for the detection of ultrasmall      how small is ultra small  ? Is this close to 40 nm...How can we call them ultrasmall then?
  11. Line 384-386, Thakur et al. concludes that  ….  all tested biofluids.   This sentence is not clear at all…
  12. Exodisc and Exo Hexa disc platforms are excellent examples of how the technology advancement occurs over time. 
  13. Line 457 – what are AAO filters?
  14. Line 459 - Is this referring to exodisc BP?
  15. Line 477 - consistent with (1)?    Not sure what this means?
  16. 11 caption should depict A, B and C separately and explanation in the text should also refer to those in a way that can be understood easily. (I had similar problem with fig.8 too, which was referred earlier)
  17. Line 526-530, PDMS layer within the gold coating?  Not sure what the authors mean?

Need to rewrite this paragraph so that readers can understand it clearly.

  1. Lines 573-581 (paragraph) describing sunkara et al looks very odd here.  It should be moved to page 13, (around line 461)   where sunkara et al's work first described....

  1. Line 590, Figure of generic neural exosome…   I assume the author is referring to fig.12.   This fig. should have 12a and 12b and be related to those, while explaining in the text..  I consider fig.12 an important one....authors should explain the two schematics in this fig. well.
  2. Line 695, authors assumed that readers know what AD means…Please describe this first time.
  3. Line 704-705, Under Sinha et al …. cell culture. Need to rewrite this.
  4. What is figure X referring to in page 21 line 721. It it referring to fig.12b?? correct this….
  5. Line 726-727, The microfluidic device  … bonded to a glass coverslip.   PDMS microchannels bonded to glass??  Not sure what the authors mean??
  6. Line 837, The first chip is made of …….. microfluidic channel runs with which the raw plasma sample passes through.  This statement is quite confusing…..Not clear at all.

Overall, It’s a great topic and authors have covered all the most important contributions in this area in the recent past.  However, I feel authors should carefully go through the manuscript, in particular, with descriptions of   microfluidic devices and detection techniques and correct the errors.   I see those throughout the manuscript.   I strongly recommend for publication of this paper after a careful and thorough revision by the authors. 

Author Response

Reviewer 4

Comments on the paper “Microfluidic Biosensing, Detection, and Profiling of Exosomes for Aging and Neurological Diseases” by   Jonalyn DeCastro et al

 In this revision, as per all the reviewer suggestions, we have altered the manuscript to focus on the main thesis of high throughput screening mechanisms for exosomes and applications for exosome screening and sensing for aging. In following this thesis, we have a more focused view on a range of microfluidic technologies for exosome isolation and detection which has subsequently condensed the manuscript to within 6700 words. Furthermore, to balance the types of technologies discussed, new microfluidic platforms were introduced with detailed descriptions of device design with advantages and disadvantages of those designs. After careful consideration and to accommodate this more diversified profile of techniques, some of the previous techniques that received reviewer comments have been removed. We believe that in doing so, this has improved the manuscript and has addressed reviewer concerns. We thoroughly inspected all figures to have detailed in-text descriptions and citations to be consistent throughout the manuscript. We additionally incorporated a new section for microfluidic antigen detection of neurodegenerative biomarkers and our future perspective of how they can be applied to exosome research. Finally, we discussed improvements required for these devices to follow through with future commercialization. We thank the reviewers for their thoughtful comments and questions about the structure, organization, balance, and device details of our manuscript.  We hope that we were able to address their questions and concerns with the modifications incorporated into this resubmission.

Topic of this review is quite interesting and timely.   I am very much impressed with the depth and breadth of coverage of the bio sensing aspects and the discussion on the limits of detection in each method.  There are two important elements in this review, while one is the bioanalysis and medical related, and the other one is microfluidic platforms and exosome detection techniques, which authors balanced well in their manuscript.    However, microfluidics aspects of the review require a bit more attention and this can improve the quality and usefulness of the paper further.  Following are some of my specific comments.

  1. Not sure, who is the author from University of California, Berkeley.  Affiliation is given but no author’s name is related to this.

We thank the reviewer for their comments. We would like to clarify the mistake in affiliation. In this resubmission, lines 6 and 10 depict the University of California, Berkeley affiliation with Michael J. Conboy, Irina M. Conboy, and Kiana Aran.

  1. Line 59-60, The device is comprised of a simple polydimethylsiloxane  (PDMS) microchannel …. field (Figure 3B).  Something missing here. 

We thank the reviewer for their comments. We apologize for the mistake in our incomplete discussion of the technology. However, in addressing a comment made by Reviewer 2 on the balance of different techniques that were covered, the discussion of Jeon et al. pressure-driven flow-induced electrophoresis device was removed from the manuscript for this resubmission. Thus, no formal discussion on the PDMS layer and its interaction with the electrical field were included in the text of this resubmission.

  1. Line 75, “…..toward the two-sided  ”  Does this mean  to say “ towards the outlets on either side of the middle outlet?

We thank the reviewer for their comments. We apologize for the confusion in the discussion of Liu et al. 2017 viscoelastic microfluidic device. However, in addressing a comment made by Reviewer 2 on the balance of different techniques that were covered and to draw a more centralized focus on exosome isolation and detection from biofluids, the discussion of Liu et al. device was removed from the manuscript for this resubmission. Thus, no formal discussion on the inlets and outlets of the viscoelastic device were included in the text of this resubmission.

  1. Line 169-170, “The detection aspect…detection area”  This sentence require some rewriting to make it clear.

We thank the reviewer for their comments. We apologize for the confusion in the discussion of Dudani et al. 2015 rapid inertial solution exchange (RInSE) microfluidic device. In this resubmission, we have now given a more in-depth discussion of the individual components of how RInSE facilitates exosome isolation in a step-by-step fashion as it follows Figure 3A i and ii. The detection aspect discussion was additionally moved to Supplemental Table 2 for a comparison of the device’s specifications.

  1. Line 178, Not sure, what figure XA referring to? 

We thank the reviewer for their comments. We apologize for the mislabeling of the figure for the description of Xia et al. 2017 single-walled carbon nanotube discussion. In this resubmission, Figure XA is Figure 4A i and ii and Figure 4B i and ii for describing the mechanism of aptamer oxidation and exosome adsorption within lines 245-257.

  1. Line 234-236, Not sure what the figure 1a and figure. 1b referring to?   Are these referring to figure6a and 6b respectively?

We thank the reviewer for their comments. We apologize for the mislabeling of the figure for the description of Tsang et al. 2019 electrochemical biosensing with graphene discussion. Figure 1a and 1b should have been referring to Figure 6a and 6b. However, in this resubmission, the discussion of Tsang et al. 2019 is under Figure 5A and B within lines 293-304.

  1. Line 248-250, The nano interfaced…detection.. This sentence is not clear…..

We thank the reviewer for their comments. We apologize for our incomplete discussion of the technology. However, in addressing a comment made by Reviewer 2 on the balance of different techniques that were covered, the discussion of Zhang et al. 2016 nano-IMEX was removed from the manuscript for this resubmission. Thus, no formal discussion on graphene oxide/polydopamine nanointerface was included the in the text for this resubmission.

  1. Line 307-308, Language correction

We thank the reviewer for their comments. We apologize for our description within lines 307 and 308 of the previous manuscript. However, in addressing a comment made by Reviewer 2 on the balance of different techniques that were covered, the discussion of Zhang et al. 2016 nano-IMEX was removed from the manuscript for this resubmission. Thus, no formal discussion on graphene oxide/polydopamine nanointerface was included the in the text for this resubmission.

  1. Many figure explanations  (in the text) were not relating the figure details well. For example,  8A and 8B are not explained well in the text.  With fig. 8B,  Authors should explain device 1 and device II parts and how the device design helped to separate exosomes by size. Otherwise, providing the figure doesn't help.

We thank the reviewer for their comments. We apologize for not giving detailed explanations in the text for the figures provided for each microfluidic technology or exosome mechanism. In this resubmission, we have re-formatted the figures and broken them up into sub-components of the figure that are individually addressed within the text. The Figure sub-components i.e.: Figure 8A and Figure 8B in the previous submission is now (lines 245-273): Figure 4A i and ii, Figure 4B i – iv, and Figure 4C i – iii with in-text, step-by-step descriptions on how these device components aid with exosome isolaton or detection.

  1. Line 364-366, A polyethylene glycol film to promote particle scattering and directs light toward the CMOS sensor, allowing for the detection of ultrasmall      how small is ultra small  ? Is this close to 40 nm...How can we call them ultrasmall then?

We thank the reviewer for their comments. We apologize for misleading description of McLeod et al. 2015 the previous manuscript. However, in addressing a comment made by Reviewer 2 on the balance of different techniques that were covered, the discussion of McLeod et al. 2015 discussion of the complementary metal-oxide semiconductor (CMOS) image sensor was removed from the manuscript for this resubmission. Thus, no formal discussion on CMOS sensor and its detection of nanoparticles was included the in the text for this resubmission.

  1. Line 384-386, Thakur et al. concludes that  ….  all tested biofluids.   This sentence is not clear at all…

We thank the reviewer for their comments. We apologize for the confusion with Thakur et al. 2017 localized surface plasmon resonance device. This statement was originally included to state that the LOD and sensitivity of the device was optimized for all the biofluids tested such as the cancer cell culture media and the serum from cancerous mice. However, in this resubmission to additionally address another reviewer’s comments on adding perspectives on the advantages and disadvantages of the design of the devices, this line was removed and inserted into the microfluidic device specification’s table (Supplementary Table 2).

  1. Exodisc and Exo Hexa disc platforms are excellent examples of how the technology advancement occurs over time. 
  2. Line 457 – what are AAO filters?
  3. Line 459 - Is this referring to exodisc BP?

We thank the reviewer for their comments. Given that comments 12-14 are all related to these devices, we would like to address these comments as a group. We thank the reviewer for highlighting the Exodisc and Exo-Hexa disc platforms as excellent examples. However, in addressing a comment made by Reviewer 2 on the balance of different techniques that were covered, the discussion of the Exodisc, Exo-Hexa disc, and Exodisc BP were removed from the manuscript for this resubmission since they only focus on size-based isolations from different biofluids. Thus, no formal discussion their filtering processes were included the in the text for this resubmission.

  1. Line 477 - consistent with (1)?    Not sure what this means?

We thank the reviewer for their comments. We would like to apologize for the inconsistencies of the citation format, in-text and within the bibliography. We have reformatted all citations to be consistent with a subscript if in-text and brackets if within a table. Citation format in the bibliography with author names, publications, years, etc. have also been reformatted for this resubmission. Line 477 “consistent with 1” is now line 346 “consistent with previous studies” and citation number 37 for Thery et al. 2009.

  1. 11 caption should depict A, B and C separately and explanation in the text should also refer to those in a way that can be understood easily. (I had similar problem with fig.8 too, which was referred earlier)

We thank the reviewer for their comments. We apologize for not giving detailed explanations in the text for the figures provided for each microfluidic technology or exosome mechanism. In this resubmission, we have re-formatted the figures and broken them up into sub-components of the figure that are individually addressed within the text. The Figure sub-components i.e.: Figure 11 A-C in the previous submission are now (lines 383-396): Figure 7A i and ii, Figure 7B with in-text, step-by-step descriptions on how these device components aid with exosome detection and exosomal mRNA profiling with machine learning.

  1. Line 526-530, PDMS layer within the gold coating?  Not sure what the authors mean?

We thank the reviewer for their comments. We apologize for misleading description of Liu et al. 2018 in the previous manuscript. However, in addressing a comment made by Reviewer 2 on the balance of different techniques that were covered, the discussion of Liu et al. 2018 surface plasmon resonance biosensor was removed from the manuscript for this resubmission. Thus, no formal discussion on the gold coating of the sensing region was included in the text for this resubmission.

Need to rewrite this paragraph so that readers can understand it clearly.

  1. Lines 573-581 (paragraph) describing sunkara et al looks very odd here.  It should be moved to page 13, (around line 461)   where sunkara et al's work first described....

 We thank the reviewer for their comments. We apologize for disorganized description of Sunkara et al. 2019 in the previous manuscript. However, in addressing a comment made by Reviewer 2 on the balance of different techniques that were covered, the discussion of Sunkara et al. 2019 Exodisc-B/P was removed from the manuscript for this resubmission. Thus, no formal discussion on the it’s methods for filtering and profiling exosomes from prostate cancer cell lines was included in the text for this resubmission.

  1. Line 590, Figure of generic neural exosome…   I assume the author is referring to fig.12.   This fig. should have 12a and 12b and be related to those, while explaining in the text..  I consider fig.12 an important one....authors should explain the two schematics in this fig. well.

We thank the reviewer for their comments. We apologize for not giving detailed explanations in the text for the figures provided for Figure 11 the explanation of the blood-brain barrier and a generic neural exosome. In this resubmission, we have re-formatted the figures and broken them up into sub-components of the figure that are individually addressed within the text. The Figure sub-components i.e.: Figure 11 in the previous submission are now (lines 616-623): Figure 11A and B with in-text descriptions.

  1. Line 695, authors assumed that readers know what AD means…Please describe this first time.

We thank the reviewer for raising this point. We would like to apologize for not reporting appropriate definitions of acronyms when first introduced in the text. In the previous work, line 695 where AD was first introduced within the main section of the manuscript, is now defined as Alzheimer’s disease (AD) line 465 in this resubmission.

  1. Line 704-705, Under Sinha et al …. cell culture. Need to rewrite this.

We thank the reviewer for raising this point. We apologize for the confusing description of Sinha et al. 2018 description of exosomal transport of neurotoxic oligomer amyloid-beta. We have re-written this line to convey that this study was performed to assess the exosome’s ability to transmit neurotoxic oligomer amyloid-beta to nearby health neurons within a cell culture model (lines 631-638 in this resubmission).

  1. What is figure X referring to in page 21 line 721. It it referring to fig.12b?? correct this….

We thank the reviewer for raising this point. We apologize for the mislabeling of figures. Figure X was referring to Figure 12B description of a generic neural exosome in the previous submission. However, in this resubmission, the discussion of this neural exosome is now Figure 11B within lines 616-619.

  1. Line 726-727, The microfluidic device  … bonded to a glass coverslip.   PDMS microchannels bonded to glass??  Not sure what the authors mean??

We thank the reviewer for their comments. We apologize for misleading description of Oh et al. 2016 in the previous manuscript. However, in addressing a comment made by Reviewer 2 on the balance of different techniques that were covered, the discussion of Oh et al. 2016 microfluidic device for monitoring the neurogenic potential of exosomes was removed from the manuscript for this resubmission. Given that the main thesis of the manuscript is exosome isolation and detection and exosome aging profiling, we have decided to remove therapeutic aspects of exosomes from the manuscript.

  1. Line 837, The first chip is made of …….. microfluidic channel runs with which the raw plasma sample passes through.  This statement is quite confusing…..Not clear at all.

We thank the reviewer for their comments. We apologize for the confusion in Ramshani et al. 2019 device. In this resubmission, we have rewritten the description as a “polycarbonate microchannel, which the raw plasma sample passes through, laid atop a piezoelectric substrate” with sub-component labeling of Figure 8 i-iv so a detailed, in-text description of all the device components for exosome lysing and miRNA profiling.

Overall, It’s a great topic and authors have covered all the most important contributions in this area in the recent past.  However, I feel authors should carefully go through the manuscript, in particular, with descriptions of   microfluidic devices and detection techniques and correct the errors.   I see those throughout the manuscript.   I strongly recommend for publication of this paper after a careful and thorough revision by the authors. 

We thank the reviewers for all their thoughtful comments and concerns. We hope we were able to address them appropriately in this resubmission and believe that with their comment, this resubmission has been greatly improved.

Round 2

Reviewer 2 Report

The MS has changed and amened as suggested. Thus, it is ready to be published.